# The Synergistic Effect of Terminal and Pendant Fluoroalkyl Segments on Properties of Polyurethane Latex and Its Film

**Fuquan Deng** [1,2], **Shuangyi Qin** [1], **Na Liu** [1] **and Wei Xu** [2,3,*]

[1] College of Art and Design, Shaanxi University of Science and Technology, Xi'an 710021, China
[2] Zhejiang Wenzhou Research Institute of Light Industry, Wenzhou 325003, China
[3] College of Bioresources Chemical and Materials Engineering, Shaanxi University of Science and Technology, Xi'an 710021, China
[*] Correspondence: xwforward@163.com; Tel.: +86-29-86168235

**Abstract:** The hydrophobic modification effect and an appropriate cost of waterborne polyurethane are regularly pursued targets of researchers. To further enhance the hydrophobic modification effect of the terminal fluoroalkyl group and control the cost, a fluorine-containing pendant group diol (DEFA) was first synthesized by the Michael addition reaction of diethanolamine (DEOA) and dodecafluoroheptyl methacrylate (G04). Next, a series of modified polyurethane latexes (TPFPU) by the terminal fluoroalkyl segments (perfluorohexyl ethanol, S104) and the pendant fluoroalkyl segments (DEFA) were synthesized by varying the DEFA dosage. Structure and performance properties of the resultants were characterized by IR, [1]H NMR, TEM, TGA, DSC, XRD, XPS, SEM, AFM and contact angle measurements. Results confirmed that the product could be successfully prepared using the present method. With the increase in DEFA dosage, the average particle size increased gradually. Thermal stability was enhanced and small regional crystals were probably produced. XPS and AFM results demonstrated that the degree of microphase separation and film roughness were increased with the increase in DEFA amount. Hydrophobicity of the TPFPU's film was also increased with the increase in DEFA dosage, and it could be guaranteed when the mass content of S104 and DEFA was larger than 17.0 wt% in total mass of raw materials, which demonstrates that the terminal and pendant fluoroalkyl groups have the favorable synergistic effect on the properties of polyurethane.

**Keywords:** fluorinated polyurethane; branched polyurethane; hydrophobicity; synergistic effect

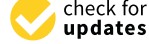



## 1. Introduction

Polyurethane film-forming materials have excellent wear resistance, oil and chemical resistance, favorable functional performance under low temperature, flexibility, good adhesion and biocompatibility, etc. [1–3]. Furthermore, they will not suppress the softness and elasticity of natural leather, endowing leather with good air permeability, corrosion resistance, friction resistance and aging resistance. Thus, more and more focus has been put on their R&D [4–8]. In recent years, due to the social advocacy of green-development, waterborne polyurethane (WPU) has gradually become a research hotspot as a coating material. However, there are many hydrophilic groups in the WPU molecular chain, resulting in poor hydrophobic properties of the film [9–11], which then contradicts its application performance. Studies have shown that the fluorine-containing carbon chain in organic fluorine compounds has a strong shielding effect and stability, and can reduce the surface energy of the material [12–15]. Therefore, organic fluorine compounds are frequently an ideal candidate to improve the hydrophobic properties of WPU films in many research works [16–24].

Until now, methods concerning fluorinated polyurethane have been mainly included as the imbedding of fluoroalkyl groups into the PU backbone as fluorine-containing small molecule diols [16,17], polyether diols [18] or macromolecular polyether diols [19,20],

the immobilizing of fluorinated segments into the PU side chain as fluorine-containing pendant groups diols [21–24], the blocking of those groups at the two PU tails [25], the distributing of fluorocarbon segments in the hyperbranched PU terminals [26–30], etc. However, the effectiveness of hydrophobic modification was unsatisfactory when using the fluorine-containing small diols, for example, octafluorohexyl diol, where the static water contact angle on its latex films could only attain 97.2° [16]. Although the contact angle of the UV-cured polyurethane acrylates with fluorine-containing macromolecular polyether (PFPE) diol could increase to 107.1° [20], the cost was not cost-effective due to numerous introductions of fluorine-containing segments. Thus, it is of great importance to more effectively improve the water resistance of waterborne polyurethane. The effectiveness of hydrophobic modification is seemingly not well guaranteed after the first three types of fluorinated PUs have been applied onto the matrices due to the disadvantage of the migration of fluoroalkyl groups in their PU molecules into the film surface [31]. In addition, research works highlight that the fluoroalkyl groups are prone to be enriched at the film surface when they are located at the polymer tails and, thus, the hydrophobicity of those polymers is more favorable [29,32,33]. Hereby, our research team had prepared novel anionic and cationic fluoroalkyl-terminated hyperbranched polyurethane (HBPUF and CHBPUF) [26,27] latexes using a hydroxyl-terminated hyperbranched polymer as a core, a polyurethane prepolymer as a bridge and perfluorohexyl ethyl alcohol as an end-capping reagent. Finally, the water resistance of the PU films could be efficiently enhanced at a lower fluorine amount. The one to three generations of hyperbranched polyesters (HBPE-1/2/3) were first synthesized by us from trimethylolpropane and 2,2-dimethylol propionic acid and then utilized to investigate their effect on the performance properties of fluoroalkyl-end-capped hyperbranched polyurethanes (FHBPU) latexes [28]. As a consequence, the increase in crosslinking density has been found to be a greater contribution to the hydrophobicity of the latex films.

However, from the perspective of the molecular structure of fluorinated branched polyurethane, hydrophobicity of the treated substrate may not acquire the due aim since the fluorine-containing end groups are not densely enough arranged on the film surface and the shielding effect is insufficient on the entire polymer substrate. Furthermore, due to the expensive price of fluorine monomer [34–36], the cost will rise if only to enhance the effect of modification and blindly increase its dosage.

Taking the above into account, the terminal and pendant fluoroalkyl segments were synchronously introduced into the PU molecular chains which were then crosslinked by the trimethylolpropane. A series of fluorinated PU latexes was accordingly synthesized to study their synergistic effect on properties of polyurethane latex and its film.

## 2. Materials and Methods

### 2.1. Materials

Trimethylolpropane (TMP), analytically pure grade, was obtained from Tianjin Guangfu Fine Chemical Co., Ltd., Tianjin, China. P-toluenesulfonic acid (p-TSA) was analytically pure and purchased from Tianjin Komeo Chemical Reagent Co., Ltd., Tianjin, China. Isophorone diisocyanate (IPDI) and 2,2-dimethylol propionic acid (DMPA), analytically pure grade, were purchased from Jining Baiyi Chemical Co., Ltd., Jining, China. Poly-butylene glycol adipate (CMA-1044) with an average molecular weight of 1000 g/mol was obtained from Yantai Huada Chemical Group Co., Ltd., Yantai, China. Triethylamine (TEA), 1,4-butanediol (BDO), acetone and dibutyltin dilaurate (DBTDL) were all analytical reagents and provided by Tianjin Fuchen Chemical Reagent Co., Ltd., Tianjin, China. Perfluorohexyl ethanol (S104) and dodecafluoroheptyl methacrylate (G04), an industrial product, were supplied by Harbin Xuejia Fluorosilicone Chemical Co., Ltd., Harbin, China. Diethanol amine (DEOA), of analytically pure grade, was purchased from Shanghai Aladdin Biochemical Technology Co., Ltd., Shanghai, China. Silicon wafers were acquired from Shanghai Songjiang Silicon Material Co., Ltd., Shanghai, China, rinsed and disposed according to our previous works [26–28].

### 2.2. Preparation of the Fluorine-Containing Chain Extender (DEFA) Based on Michael Addition Reaction between DEOA and Fluorinated Acrylate Monomer (FA, G04)

First, 2.21 g of DEOA, 0.92 g sodium ethanol and 50 mL of anhydrous ethanol were added into a three-neck flask equipped with a thermometer, a reflux condenser and a nitrogen catheter. Then, 8 g of G04 was dropped slowly into it with gentle stirring under ice cooling, and the reaction continued at 60 °C for 24 h under an $N_2$ atmosphere. After the reaction, the system was cooled down to room temperature and anhydrous ethanol was removed by reduced pressure distillation, the fluorine-containing chain extender (DEFA) was prepared by repeated extraction using deionized water to remove the unreacted DEA and sodium alcohol, and reduced pressure distillation was used to remove the unreacted G04. Scheme 1 here illustrates the entire reaction process of the pendant fluorine-containing chain extender (DEFA).

**Scheme 1.** Schematic diagram of the TPFPU latexes preparation.

### 2.3. Preparation of the Terminal and Pendant Fluoroalkyl Segments Co-Modified Polyurethane (TPFPU) Latexes

The stoichiometric IPDI and CMA-1044 were first added into the 250 mL three-necked flask, and nitrogen gases were bubbled into it for 0.5 h. Next, the reaction mixture was heated to 80 °C with vigorous stirring. Three drops of DBTDL were dripped into the system as a catalyst, and the reaction was maintained for 1 h. Then, a portion of DEFA, BDO and

DMPA was added to the mixture, and the chain extension reaction was maintained for 2.5 h. At this time, NCO-terminated fluorine-containing polyurethane prepolymer was acquired. After this, acetone solutions containing the given S104 and TMP were successively dripped into the above system and reacted separately for 2 h. Then, the neutralization reaction proceeded for 0.5 h at 40 °C with TEA as the neutralizing agent. Shortly after this, the terminal and pendant fluoroalkyl segments co-modified polyurethane (TPFPU) was obtained with a white and viscous appearance. Finally, the TPFPU was dispersed with deionized water under vigorous stirring, and the slightly blue light TPFPU latex was acquired with about 30% of the solid content. Scheme 1 also illustrates the entire reaction process of TPFPU, and Table 1 lists the detailed recipes of the TPFPU latexes preparation.

**Table 1.** The preparation formula for the TPFPUs.

| Samples | TMP (g, mmol) | IPDI (g, mmol) | CMA-1044 (g, mmol) | DMPA (g, mmol) | BDO (g, mmol) | DEFA (g, mmol) | S104 (g, mmol) |
|---------|---------------|----------------|--------------------|----------------|---------------|----------------|----------------|
| TPFPU-1 | 0.16, 1.17 | 11.11, 50 | 18.50, 18.5 | 1.07, 8 | 1.08, 12 | 0.00 | 1.28, 3.5 |
| TPFPU-2 | 0.16, 1.17 | 11.11, 50 | 18.50, 18.5 | 1.07, 8 | 0.90, 10 | 1.01, 2 | 1.28, 3.5 |
| TPFPU-3 | 0.16, 1.17 | 11.11, 50 | 18.50, 18.5 | 1.07, 8 | 0.72, 8 | 2.02, 4 | 1.28, 3.5 |
| TPFPU-4 | 0.16, 1.17 | 11.11, 50 | 18.50, 18.5 | 1.07, 8 | 0.54, 6 | 3.03, 6 | 1.28, 3.5 |
| TPFPU-5 | 0.16, 1.17 | 11.11, 50 | 18.50, 18.5 | 1.07, 8 | 0.36, 4 | 4.04, 8 | 1.28, 3.5 |
| TPFPU-6 | 0.16, 1.17 | 11.11, 50 | 18.50, 18.5 | 1.07, 8 | 0.18, 2 | 5.05, 10 | 1.28, 3.5 |
| TPFPU-7 | 0.16, 1.17 | 11.11, 50 | 18.50, 18.5 | 1.07, 8 | 0.00 | 6.06, 12 | 1.28, 3.5 |

### 2.4. Formation of the TPFPU Latex Film

About 30 g of the TPFPU emulsion was poured onto PTFE plates and dried at an ambient temperature for 72 h to form the TPFPU films. Next, those TPFPU films were dried at 65 °C for 24 h in an oven. After cooling, those films were gently peeled off from the PTFE plates and washed consecutively with anhydrous ethanol and deionized water several times. Finally, the latex films were dried in a vacuum drying chamber for later use.

### 2.5. Characterization

An FTIR spectrometer (VECTOR-70, Bruker Ltd., Bremen, Germany) was used to characterize the FTIR spectra of the samples in the range of 500–4000 cm$^{-1}$. The scanning frequency and resolution were adopted as 32 times and 2 cm$^{-1}$, respectively. The raw materials, DEOA and G04, and the purified chain extender, DEFA, were determined with the KBr coating method, and the several TPFPU polymers were measured with the KBr pellets method.

A spectrometer (INOVA-400, Varian Ltd., Palo Alto, CA, USA) was used to characterize the $^1$H NMR spectrum of TPFPU-4 with Tetramethylsilane (TMS) as an internal standard and chloroform-d as a solvent.

The particle size and size distribution of the TPFPU latexes were determined by the laser particle size analyzer (Nano-ZS, Malvern Co., Ltd., Whitnash, UK). The TPFPU latexes were diluted to a suitable extent and then stained with 2 wt% phosphotungstic acid liquor. Next, the TPFPU latex particles were observed and some photographs were taken by the Transmission Electron Microscope (TEM, Tecnai G2 F20 S, FEI. Co., Ltd., Hillsboro, OR, USA).

The thermal stability of the sample was analyzed by a thermogravimeter (Q500, TA Co., Ltd., New Castle, DE, USA), where the heating rate of 10 °C/min was applied under the inert atmosphere and the temperature range was from the ambient temperature to 600 °C.

A thermal analyzer (Q2000, TA Ltd., New Castle, DE, USA) was used to conduct the DSC experiments with the heating rate of 10 °C/min in the temperature range of −60 °C to 150 °C.

Raw material CMA-1044 and several TPFPU polymers were placed in the groove of the sample plate. Then, they were tested with an X-ray diffractometer (D8 Advanced, Bruker

Ltd., Bremen, Germany) where $CuK_{\alpha}$ radiation ($\alpha = 0.154$ nm) was used with generator settings of 45 mA and 45 kV. The diffraction angle range was 5~60°, the diffraction step size was 0.02°, and the diffraction step was 0.1s in continuous mode.

The determination on the sample elements was implemented by an X-ray photoelectron spectrometer (Axis Supra, Kratos Analytical Co., Ltd., Stretford, UK) in which the monochromatic Al $K_{\alpha}$ rays were utilized, as the X-ray source, angle of the incidence, and the vacuum value in the analysis room were 90° and $1.2 \times 10^{-8}$ Pa, respectively. The spectra line deviation was rectified using the binding energy value (284.8 eV) of the contaminated carbon $C_{1s}$ on the sample surface.

A scanning electron microscope (VEGA3, TESCAN Ltd., Brno, Czech Republic; magnification: 2500 times; acceleration voltage: 10.0 kV) was utilized to observe the surface morphologies of the TPFPU films. All samples were anchored onto the carrier table using a conductive adhesive and were then coated with gold under a vacuum for standby.

A SPA-400 AFM (Seiko Instruments Inc., Chiba, Japan) was used to acquire the fine morphologies of several TPFPUs on a silicon wafer as the tapping mode under the condition of $20 \pm 0.5$ °C and $49.5 \pm 2\%$ relative humidity.

Water contact angles on the TPFPU films were determined by a contact angle goniometer (OCA20, Dataphysics Ltd., Filderstadt, Germany), where 5 μL of distilled water of the injecting volume was adopted. For diminishing the experimental errors, at least nine repeated determinations were made, and their average was used as the final result for one sample. TPFPU films were immersed in distilled water at ambient temperature for 48 h and then removed from the water and drained for several seconds. While the excess waters were wiped with filter paper, the water absorbency Q could be calculated by the following formula:

$$Q = \frac{W_2 - W_1}{W_1} \times 100\%$$

where $W_2$ is the weight of the wet film with water, and $W_1$ is the weight of the dry film.

A universal testing machine (UTM2102, Sansizongheng Technology Co., Ltd., Shenzhen, China) was applied to test the mechanical properties of the TPFPU films with the extension rate of 5 mm/min at room temperature. The films were cut into dumbbell-shaped pieces (width: 3 mm, length: 10 mm).

## 3. Results and Discussion

### 3.1. Structure Characterization

FTIR spectra of the raw materials, G04 and DEOA, and the fluorine-containing chain extender (DEFA) are presented in Figure 1. It can be seen that the strong signal at 1734 $cm^{-1}$ was assigned to the carbonyl group, and the weak absorption at 1641 $cm^{-1}$ was due to the stretching vibration of alkene in Figure 1a. The wide absorption peaks between 1258 and 1100 $cm^{-1}$ originated from the overlapping between the stretching vibration of C–F and the asymmetrical stretching vibration of C–O. The absorption peak at 695 $cm^{-1}$ was ascribed to the wagging vibration of C–F. In Figure 1b, the strong absorption peak at 3477 $cm^{-1}$ and the weak absorption peak at 1650 $cm^{-1}$ resulted from the stretching vibration of –OH and the bending vibration of –NH–. However, in Figure 1c, the absorption peak of alkene at 1641 $cm^{-1}$ disappeared, and the absorption peaks of –OH at 3477 $cm^{-1}$, C=O at 1734 $cm^{-1}$, –$CH_2$– and –$CH_3$ between 2962 and 2904 $cm^{-1}$, C–O at 1242 $cm^{-1}$ and –$CF_2$– at 1258, 1169 and 695 $cm^{-1}$ existed. In brief, the above facts testify that G04 and DEOA have completed the reactions to acquire the due fluorine-containing chain extender.

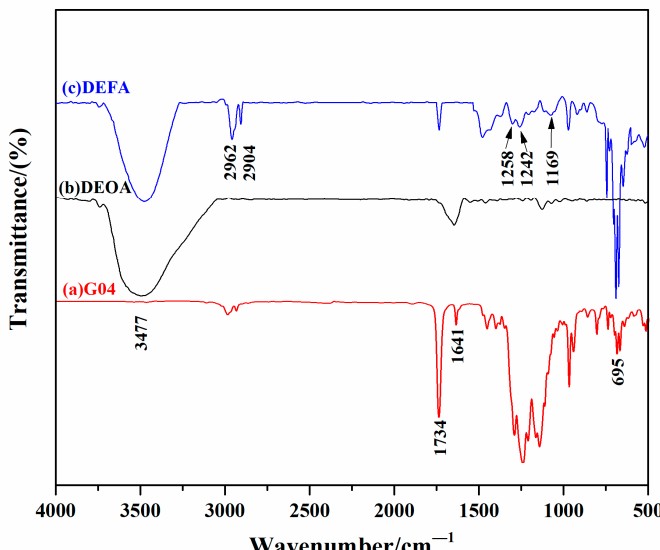

**Figure 1.** IR spectra of (**a**) fluorine-containing monomer (G04), (**b**) diethanolamine (DEOA) and (**c**) fluorinated chain extender DEFA.

FTIR spectra of several terminal and pendant fluoroalkyl segments' co-modified polyurethanes are shown in Figure 2. The absorption peaks at 3311 cm$^{-1}$ and 1538 cm$^{-1}$ were assigned to the stretching vibration and bending vibration of N–H in urethane groups, and the absorption peaks at 1716 cm$^{-1}$ and 1404 cm$^{-1}$ resulted from the signals of C=O and C–N in urethane groups [28]. The absorption peaks at 1244 cm$^{-1}$ and 1017 cm$^{-1}$ were derived from the stretching vibrations of C–O and C–O–C, and the absorption peaks at 1171 cm$^{-1}$ and 730 cm$^{-1}$ were obtained from the stretching vibration and bending vibration of –CF [28]. In addition, these signals from the fluorocarbon groups were intensified with the increasing addition amount of DEFA into the TPFPU backbones. Those results indicate that the DEFA has been successfully combined into the PU chains.

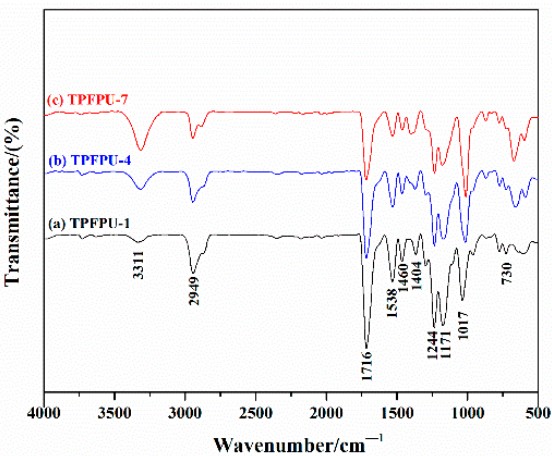

**Figure 2.** IR spectra of (**a**) TPFPU-1, (**b**) TPFPU-4 and (**c**) TPFPU-7.

The TPFPU-4 was chosen as a sample to analyze the $^1$H NMR spectrum and CDCl$_3$ as a solvent. The result is revealed in Figure 3. The signals at 0.99 ppm (*b*H), 1.19 ppm (*c*H), 1.32 ppm (*d*H), 3.73 ppm (*k*H) and 2.86 ppm (*i*H) were due to the IPDI unit [37]. The signals at 1.40 ppm (*e*H), 2.05 ppm (*g*H) and 4.08 ppm (*m*H) were ascribed to the CMA-1044 unit [27]. The signals at 0.99 ppm (*b*H) and 4.08 ppm (*m*H) came from the DMPA unit. The signals at 2.86 ppm (*i*H), 3.89 ppm (*l*H) and 4.08 ppm (*m*H) originated from the DEFA unit. The signals at 1.66 ppm (*f*H), 2.30 ppm (*h*H) and 4.08 ppm (*m*H) were attributed to the BDO

and S104 units. The signals at 0.90 ppm (*a*H), 1.19 ppm (*c*H) and 3.40 ppm (*j*H) were due to the core TMP [37]. In a word, the above [1]H NMR results are in accordance with the FTIR results, and those results testify that we have successfully synthesized the terminal and pendant fluoroalkyl segments' co-modified polyurethanes via the present methods.

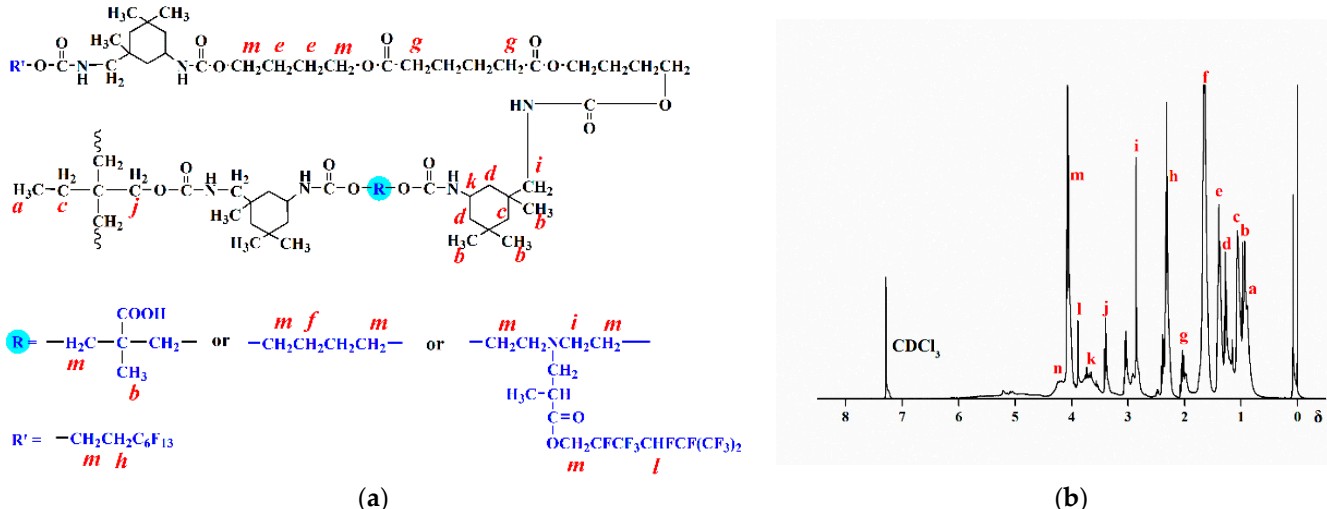

| (**a**) | (**b**) |

**Figure 3.** [1]H NMR spectrum of TPFPU-4. (**a**) labels of hydrogen protons, (**b**) [1]H NMR spectrum.

### 3.2. Particle Sizes and Morphologies of the TPFPUs

The particle sizes and morphologies of the TPFPUs were determined by nano particle size analyzer and transmission electron microscope, respectively and results are visualized in Figures 4 and 5. The average particle sizes of the TPFPU-1, TPFPU-4 and TPFPU-7 were 30.4 nm, 66.8 nm and 120.5 nm and their PDI values were 0.091, 0.152 and 0.201. The underlying reason for these PU particle size variations is probably due to the continuous increase in the fluorine-containing segments content and their interaction, which would limit the chain movement and enlarge the latex particles or be not conducive to the dispersion of latex particles in water [38,39]. Finally, the average particle sizes and the PDI indexes of the TPFPU-1, TPFPU-4 and TPFPU-7 were continually increased.

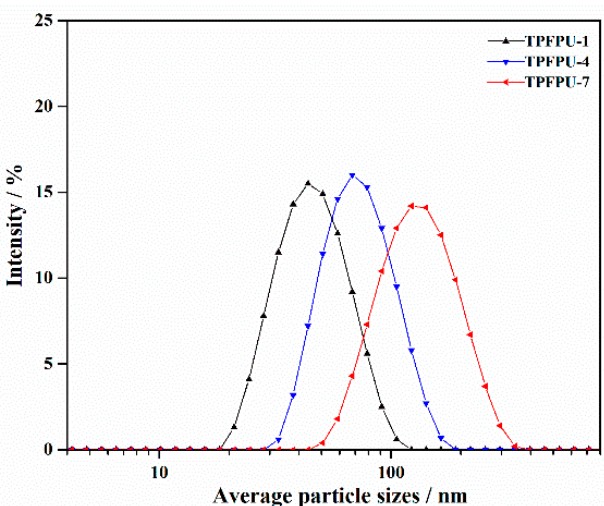

**Figure 4.** Particle size distribution of the TPFPUs.

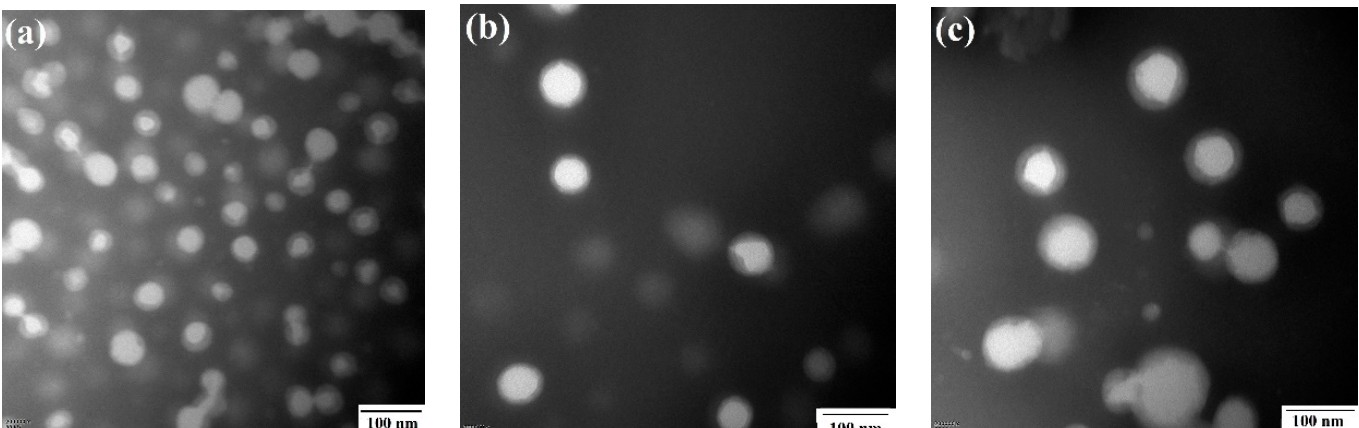

**Figure 5.** TEM of the TPFPUs: (**a**) TPFPU-1, (**b**) TPFPU-4 and (**c**) TPFPU-7.

From the TEM images of the TPFPU-1, TPFPU-4 and TPFPU-7, the approximate regular spherical morphologies of three PU particles clearly emerged, and their particle's diameters were estimated at about 20 nm, 50 nm and 110 nm according to the scale bars in TEM images. In light of the influence of the hydration layer in the DLS measurement, the results of the DLS and TEM are in good accordance with each other.

### 3.3. Thermal Behaviors of the TPFPUs

The thermal behavior of three polymers was characterized by TGA and DSC in this text, and the results are shown in Figures 6 and 7. As seen in Figure 6, the initial decomposition temperatures of the three TPFPUs were all about 175 °C, and weight loss at this stage may result from impurities or small molecular products. Their $T_{d50\%}$ was gradually enhanced and attained 291 °C, 307 °C and 335 °C, respectively, which demonstrates that thermal stability of the TPFPU-1, TPFPU-4 and TPFPU-7 is consecutively strengthened. Those consequences should be due to the gradual shielding effect of the fluorinated segments with high stable bond energy [12–15].

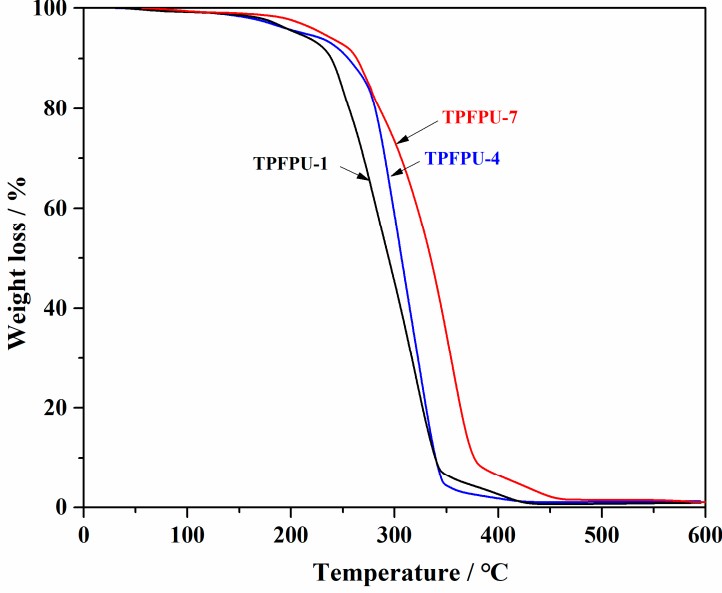

**Figure 6.** TGA curves of the TPFLUs.

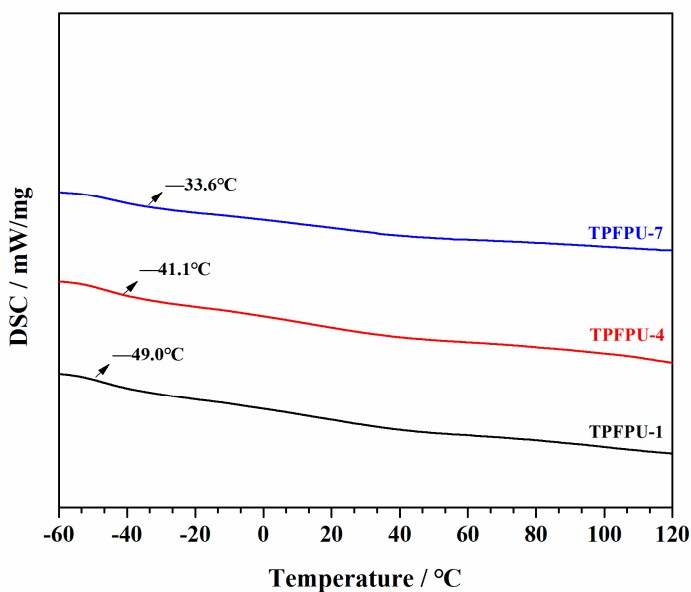

**Figure 7.** DSC curves of the TPFPUs.

There was only one $T_g$ in all the TPFPUs curves, and these were $-49.0$ °C, $-41.1$ °C and $-33.6$ °C in the TPFPU-1, TPFPU-4 and TPFPU-7, which were all below zero °C and should be derived from the soft polyester segments. With the ever-increasing fluorine-containing segments' content, their shielding protections on the molecular chains, including the soft polyester segments, are gradually enhanced, and the ability of molecular chains to resist thermal motion is improved gradually. Thus, $T_g$ of the soft polyester segments in the TPFPU-1, TPFPU-4 and TPFPU-7 is enhanced. The obtained results are similar to the research in [26,27], which investigated the effect of the fluoroalkyl tails with strong polarity on the thermal motion of the inner molecular chains in PU while the polymer was heated, and, finally, $T_g$ of polymers containing the fluoroalkyl tails was enhanced.

*3.4. XRD*

X-ray diffraction is often used to investigate the fine structure of materials, and some information such as crystal structure and amorphous form can be obtained [26–28]. Thus, the raw material, CMA-1044, and three polymer latex films of the TPFPU-1, TPFPU-4 and TPFPU-7 were tested by XRD instrument, and results are seen in Figure 8. Two distinct pinnacles were found at 2θ = 21° and 24° in the CMA-1044 XRD curve, respectively, which demonstrates that the CMA-1044 possesses the crystallinity. However, a wide diffuse peak around 2θ = 20° was observed in all TPFPUs, and there were weak sharp peaks at 2θ = 20° in the TPFPU-4 and TPFPU-7. In addition, the intensity of the sharp peaks was slightly increased with the increasing amount of the fluorinated segments. Those results indicate that the amorphous state mainly appears in the TPFPUs and the microphase separation degree is relatively low. With the increasing amount of the embedded fluorinated segments, hydrogen bonds' interactions between pendant fluorocarbon segments and imine groups from the hard segments will be gradually intensified. Next, the microphase separation degree of the TPFPU between hard segments and soft segments is enhanced, and small regional crystals are probably produced.

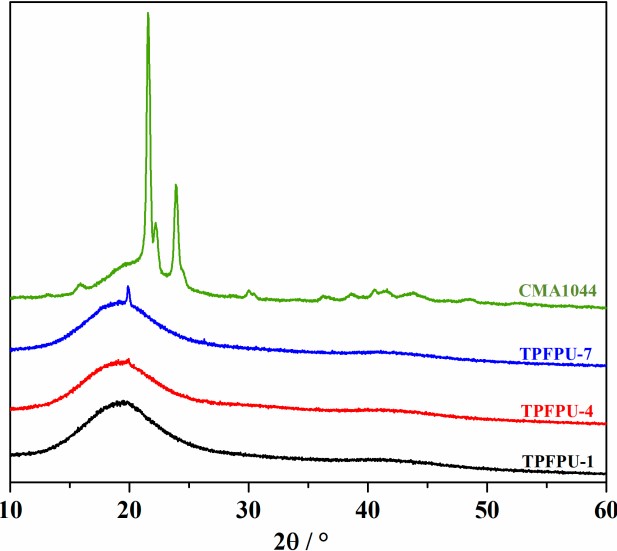

**Figure 8.** XRD pattern of the TPFPU and the CMA-1044.

### 3.5. Surface Analysis

Surface chemical element compositions of the TPFPU-1, TPFPU-4 and TPFPU-7 latex films were determined using the XPS technique, and the wide spectra are shown in Figure 9. It can be observed from Figure 9 that four peaks appeared at the binding energies of 284, 398, 531 and 686 eV, which belonged to $C_{1s}$, $N_{1s}$, $O_{1s}$ and $F_{1s}$, respectively [31]. Furthermore, the intensity of the fluorine element was gradually increased. Based on the measured results, the experimental atomic compositions of the TPFPU latex films were obained. Moreover, the theoretical atomic compositions were calculated from the experimental recipes. Their comparative results are listed in Table 2. The measured values of the F element were all larger than the corresponding theoretical ones in three TPFPU samples. Those above outcomes indicate that the fluorinated segments with low surface free energy have the tendency to migrate and aggregate to the air–film surface during the film-forming process.

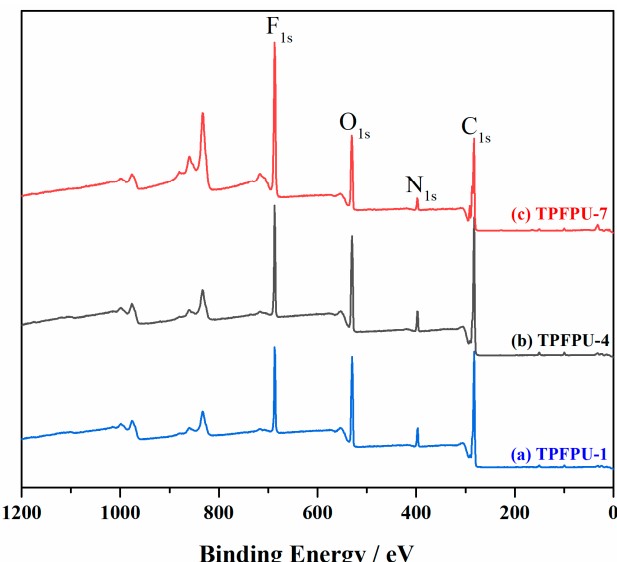

**Figure 9.** XPS survey spectra of (**a**) TPFPU-1, (**b**) TPFPU-4 and (**c**) TPFPU-7.

**Table 2.** The theoretical and experimental atomic compositions of the TPFPU latex films.

| | Element | $F_{1s}$/% (Mass) | $O_{1s}$/% (Mass) | $C_{1s}$/% (Mass) | $N_{1s}$/% (Mass) |
|---|---|---|---|---|---|
| TPFPU-1 | Experimental value | 11.14 | 17.16 | 66.86 | 4.84 |
| | Theoretical value | 7.40 | 26.16 | 62.44 | 4.00 |
| TPFPU-4 | Experimental value | 13.54 | 16.19 | 65.40 | 4.87 |
| | Theoretical value | 10.20 | 23.97 | 61.66 | 4.17 |
| TPFPU-7 | Experimental value | 17.75 | 15.08 | 61.73 | 5.44 |
| | Theoretical value | 13.00 | 21.67 | 61.07 | 4.26 |

### 3.6. Film Morphologies

A scanning electron microscope was used to observe the microstructure of three TPFPU latex films, and the results are displayed in Figure 10. Three of the TPFPU latex films were all irregular in appearance. TPFPU-1 latex film presented the stripe-like pattern. TPFPU-4 latex film was wrinkled, with some white fine particles on its surface. Additionally, TPFPU-7 latex film presented the heterogeneous morphology, with some hollows and bubble-like bulges on its surface. Those stripes or wrinkles were probably due to the inherent cohesions amongst molecular chains, which made the latex films uneven. Some white fine particles resulted from small crystal regions. With the increasing amount of the embedded fluorinated segments, the microphase separation degree of the TPFPU was enhanced and small regional crystals were probably produced, and these consequences seem to be consistent with the above XRD result.

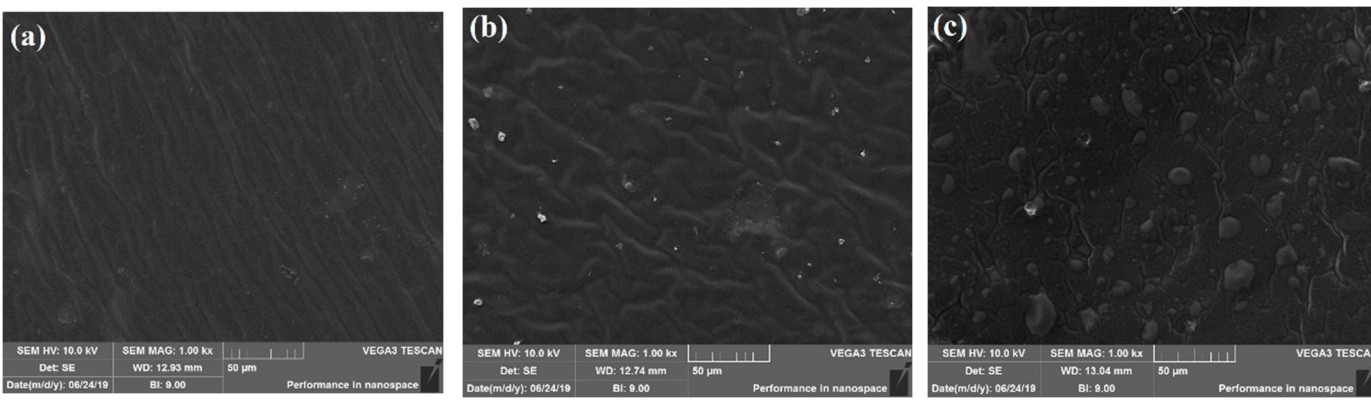

**Figure 10.** SEM images of (**a**) TPFPU-1, (**b**) TPFPU-4 and (**c**) TPFPU-7.

To obtain more fine morphologies of the TPFPUs, their morphologies on the silicon wafer were observed with the AFM instrument, and the results are presented in Figure 11. Additionally, some roughness factors such as root–mean–square roughness ($R_q$) and mean roughness ($R_a$) were also given in Table 3. Three different TPFPUs all exhibited the heterogeneous and phase-separated patterns, with some bright pinnacles or peaks on their AFM topographies. With the increasing amount of the embedded fluorinated segments, the quantity and size of the bright peaks were increased. From Table 3, it can be seen that roughness factors, such as $R_q$ and $R_a$, were in the ascending order of TPFPU-1, TPFPU-4 and TPFPU-7.

**Table 3.** Roughness parameters of the TPFPU latex films.

| Samples | $R_q$ (nm) | $R_a$ (nm) |
|---|---|---|
| TPFPU-1 | 2.651 | 1.478 |
| TPFPU-4 | 4.754 | 2.593 |
| TPFPU-7 | 5.485 | 4.086 |

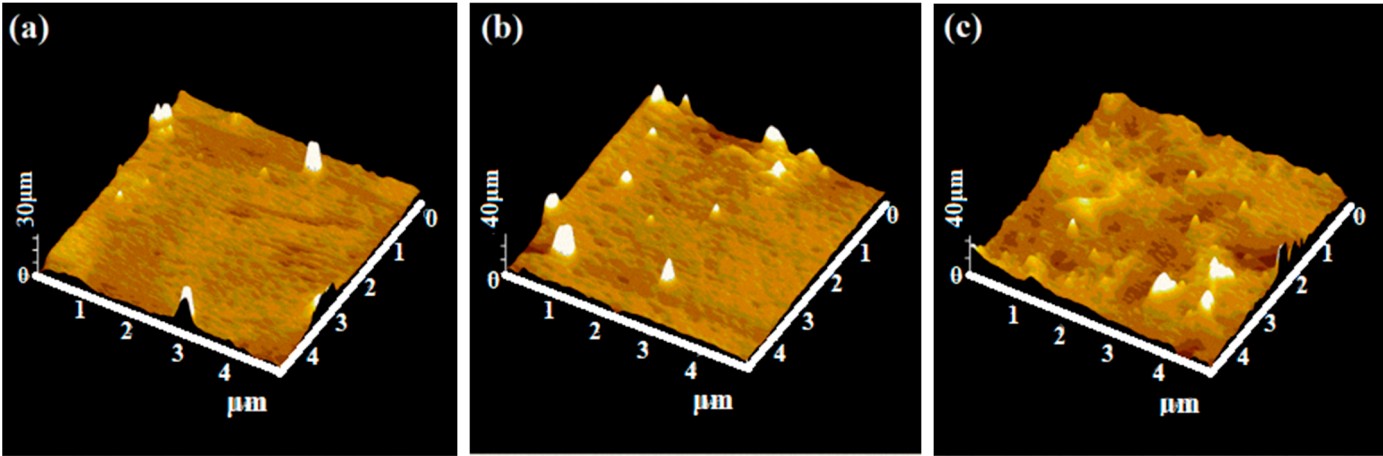

**Figure 11.** AFM topographies of (**a**) TPFPU-1, (**b**) TPFPU-4 and (**c**) TPFPU-7.

### 3.7. Hydrophobicity of the TPFPU Films

Static water contact angle and water absorptivity are two commonly used methods to study the hydrophobic properties of one film [26,27,37]. Thus, water contact angles (WCAs) on the TPFPU films and their levels of water absorption were measured, and the results are presented in Table 4. It is clearly shown that the static WCA and water absorptivity of the TPFPU-1 film, only the fluorine-containing end group (S104) modified polyurethane, were 92.8° and 94.3%. Its hydrophobicity was apparently unsatisfactory. With the increasing amount of the embedded fluorinated segments, static WCA was steadily increased and water absorptivity was decreased instead. However, this change trend almost ceased until the amount of the embedded fluorinated segments was attained to some extent. At this time, the static WCA and the water absorptivity of the TPFPU latex film were about 110° and 10%. Through calculation of theoretical mass content of DEFA and S104 or sole DEFA in total raw materials, we acquired the relevant information seen in Table 4. As mass content of DEFA and S104 in total raw materials was only 3.9 wt% at first, it was too small to be densely arranged on the film surface. Thus, the shielding effect is insufficient on the whole polymer substrate, and hydrophobicity of the TPFPU-1 was unsatisfactory. As the mass content of DEFA and S104 was steadily increased from 3.9 wt% to 17.0 wt%, that of DEFA was increased from 0 to 13.5 wt%. Accordingly, the shielding protection of fluoroalkyl segments, including the terminal and pendant ones, on the entire polyurethane molecular chains was gradually saturated. Finally, the static WCA and the water absorptivity of TPFPU latex film were almost kept at 110° and 10%, respectively. Comparing these results with other literature concerning fluorinated WPU [21–24], the same results that WCA of fluorinated WPU films increased with the fluorine content could be found. Therefore, those results in this context demonstate that the terminal and pendant fluoroalkyl groups have a favorable synergistic effect on hydrophobicity of waterborne polyurethane.

**Table 4.** Water contact angles, water absorptivity and theoretical mass content of DEFA and S104 or DEFA of the TPFPU latex films.

| Samples | Water Contact Angles (°) | Water Absorptivity (%) | Mass Content of DEFA and S104 in Total Raw Materials (wt%) | Mass Content of DEFA in Total Materials (wt%) |
|---|---|---|---|---|
| TPFPU-1 | 92.8 ± 1.2 | 94.3 | 3.9 | 0 |
| TPFPU-2 | 94.1 ± 1.2 | 87.6 | 6.7 | 3.0 |
| TPFPU-3 | 97.6 ± 1.6 | 63.5 | 9.5 | 5.8 |
| TPFPU-4 | 104.6 ± 1.9 | 49.3 | 12.1 | 8.5 |
| TPFPU-5 | 106.4 ± 1.8 | 42.5 | 14.6 | 11.1 |
| TPFPU-6 | 110.5 ± 2.3 | 10.2 | 17.0 | 13.5 |
| TPFPU-7 | 110.6 ± 2.1 | 9.9 | 19.2 | 15.9 |

### 3.8. Mechanical Properties of the TPFPU Films

Mechanical properties of the TPFPU films were tested as tensile strength and elongation at a break, and results are shown in Figure 12. With the increasing amount of the embedded fluorinated segments, tensile strength was first increased and then decreased, and elongation at a break was reduced continuously. The tensile strength of the TPFPU-5 was the greatest. The possible reason is due to the increase in hydrogen bonds or interactions amongst the PU molecular chains, which will intensify rigidity and reduce flexibility of the film. As shown in Figure 2 of a series of TPFPU FTIR, the absorption peaks of imine groups were shifted towards lower frequencies at 3311 cm$^{-1}$ and their intensities were increased with the order of TPFPU-1, TPFPU-4 and TPFPU-7, which denotes that there are hydrogen bonds in all of the TPFPUs and the degrees of hydrogen bonds are increased with the order of TPFPU-1, TPFPU-4 and TPFPU-7. Therefore, the results of mechanical properties and IR were related to each other in a certain sense. In addition, in consideration of the crosslinked structure of the present polymers, when the amount of the embedded fluorinated segments is so high that hydrogen bonds or intermolecular interactions may occur in excess, that is, the excessive cross-linking could be unfavorably engendered, this will lead to the molecular chain hardly moving freely, and the mechanical properties decrease. Those consequences are also consistent with the related literature [40,41].

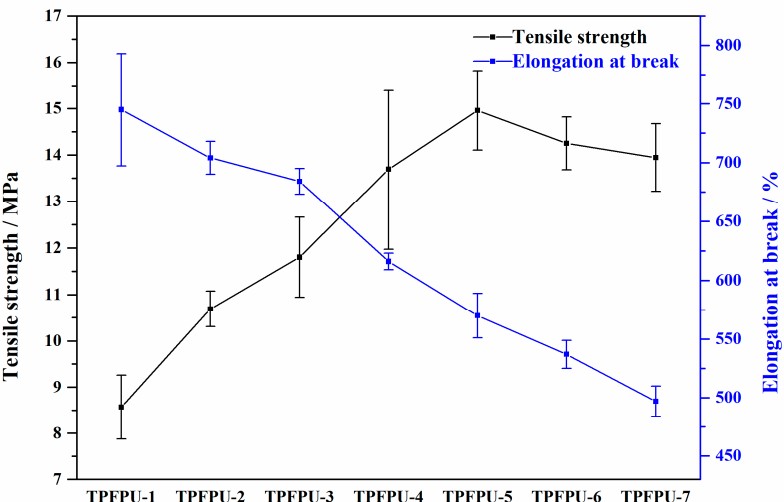

**Figure 12.** Mechanical properties of the TPFPU films.

### 4. Conclusions

A fluorine-containing chain extender (DEFA) has been synthesized by the Michael addition reaction of DEOA and G04. A series of the terminal and pendant fluoroalkyl co-modified polyurethane (TPFPU) latexes has also been prepared. Results confirmed that the product could be successfully prepared. With the increase in DEFA dosage, the average particle size increased gradually. Thermal stability was enhanced and small regional crystals were probably produced. XPS and AFM results demonstrated that the degrees of microphase separation and film roughness were increased with the increase in DEFA amount. Hydrophobicity of the TPFPUs film was also increased with the increase in DEFA dosage, and it could be guaranteed when the mass content of S104 and DEFA was larger than 17.0 wt% in total mass of raw materials, which demonstrates that the terminal and pendant fluoroalkyl groups have a favorable synergistic effect on the properties of polyurethane.

**Author Contributions:** Conceptualization, W.X. and F.D.; methodology, F.D.; formal analysis, S.Q. and N.L.; writing—original draft preparation, F.D.; writing—review and editing, W.X. All authors have read and agreed to the published version of the manuscript.

**Funding:** The authors show great appreciation to the Projects from the Key Research and Development Program of Shaanxi Province (No. 2020ZDLGY13-11), Key Program of Wenzhou (No. ZG2017028) and the Doctorial Research Foundation of Shaanxi University of Science and Technology (No. BJ13-22) for their financial support.

**Institutional Review Board Statement:** Not applicable.

**Informed Consent Statement:** Not applicable.

**Data Availability Statement:** Data sharing is not applicable to this article.

**Conflicts of Interest:** The authors declare no conflict of interest.

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
