# Peer review of "The Synergistic Effect of Terminal and Pendant Fluoroalkyl Segments on Properties of Polyurethane Latex and Its Film"

_coatings, doi:10.3390/coatings12091271_

Round 1
Reviewer 1 Report
This work is an advance based on the previous works of the authors looking for improvements on the quality of the based on fluroalkylpolurethanes. It is an interesting work that supports the increasing work in this topic.
Some comments to contribute to the improvement are included in the reviewer's file.

Author Response
Please kindly see in attached files.

Reviewer 2 Report
In this study, authors prepared polyurethane latex and films and evaluated their different properties. I recommend this paper for publication but urge the authors to consider the following points seriously during the revision.
Abstract is not scientifically sound. It should begin with rationale.
Identify the problem through literature review.
Aim and objectives of the work are also not clearly mentioned in the present paper.
Authors should explain what new and original this paper has to offer beyond to the already existing in the literature.
There are some grammatical mistakes in manuscript. Please revise it carefully.
Please cite some relevant latest papers of the respective journal.
Author Response
Please see it in attached files.

Reviewer 3 Report
The authors demonstrated the effect of terminal and pendant fluoroalkyl segments on the thermal, mechanical, and surface properties. The authors utilized a DEFA as a fluorine-containing chain extender in the waterborne polyurethane to improve the thermal, mechanical, and surface properties. The authors explained that the thermal, mechanical, and surface properties of TPFPUs are improved when the DEFA concentration increases. Although the authors introduced a DEFA into the waterborne polyurethane, there are not enough scientific discussion on all results. In addition, the written English should be double-checked.
- In the Introduction Section, the authors should introduce a variety of polyurethane properties including most-cited references as well as the surface properties of waterborne polyurethane in detail.
- What is the novelty of this work compared to the references of 9 and 10 ? The concept the polymer composition to use the fluorinated segments into the PU side chain is similar to that in this work.
- The authors explained that the PU particle size increases with the increase of DEFA amounts. The discussion to increase the particle size does not make sense. In the references of 18 and 19, there is no explanation of poor emulsifying effect. In addition, the authors should not reckon but measure the particle size with the error range from the TEM images.
- What is the TEM mode? Why do the particles look brighter one?
- With the increase of DEFA amounts, the crystallinity of TPFPU increased. Please explain why and also the schematic illustration including the related references. In addition, why did the crystalline peaks of CMA-1044 disappear?
- The authors should provide the AFM images in broader area that can compare to the SEM results.
- The hydrophobicity increased with the increase of DEFA amounts because of the effect of the terminal and pendant fluoroalkyl groups, which is lower than that in the reference of 9. Please explain the merit of this work compared to the other fluorinated WPU. Also, is there any roughness effect on the improved hydrophobicity in this work?
- TPFPU-5 exhibited the highest tensile strength and the authors explained that it is because “when amount of the embedded fluorinated segments is so much that small regional crystals are probably produced and those inhomogeneous structures will be detrimental to tensile strength of the film.” The heterogeneous (not inhomogeneous) structures by microphase separation will increase the mechanical properties, but not in this work. Please explain why including the schematic illustration in detail.
- The quality of Figure 5 should be improved including the scale bar.
Based on the above consideration, the reviewer can recommend to publish the article in this journal after major revision.
Author Response
Please see it in attached files.

Round 2
Reviewer 2 Report
Comments are addressed adequately. Now manuscript can be accepted.
Reviewer 3 Report
The authors revised the manuscript according to the reviewer's comment.